Subject Areas:
environmental science/computational biology/mathematical modelling

Keywords:
Sirius crop simulation model, local-scale climate scenarios, LARS-WG weather generator, drought and heat stress

Author for correspondence:
Thibaut Putelat
e-mail: thibaut.putelat@rothamsted.ac.uk

# Local impacts of climate change on winter wheat in Great Britain

Thibaut Putelat[1,3], Andrew P. Whitmore[1], Nimai Senapati[2] and Mikhail A. Semenov[2]

[1]Department of Sustainable Agriculture Sciences, Rothamsted Research, Harpenden AL5 2JQ, UK
[2]Department of Plant Sciences, Rothamsted Research, Harpenden AL5 2JQ, UK
[3]Department of Engineering Mathematics, University of Bristol, Bristol, BS8 1UB, UK

  TP, 0000-0002-0158-033X; APW, 0000-0001-8984-1436;
NS, 0000-0002-0238-6694; MAS, 0000-0002-1561-7113

Under future CMIP5 climate change scenarios for 2050, an increase in wheat yield of about 10% is predicted in Great Britain (GB) as a result of the combined effect of $CO_2$ fertilization and a shift in phenology. Compared to the present day, crops escape increases in the climate impacts of drought and heat stresses on grain yield by developing before these stresses can occur. In the future, yield losses from water stress over a growing season will remain about the same across Great Britain with losses reaching around 20% of potential yield, while losses from drought around flowering will decrease and account for about 9% of water limited yield. Yield losses from heat stress around flowering will remain negligible in the future. These conclusions are drawn from a modelling study based on the response of the Sirius wheat simulation model to local-scale 2050-climate scenarios derived from 19 Global Climate Models from the CMIP5 ensemble at 25 locations representing current or potential wheat-growing areas in GB. However, depending on susceptibility to water stress, substantial interannual yield variation between locations is predicted, in some cases suggesting low wheat yield stability. For this reason, local-scale studies should be performed to evaluate uncertainties in yield prediction related to future weather patterns.

## 1. Introduction

Wheat production in the 2050s is likely to face conflicting pressures. Firstly, up to 70% more food will be needed to feed the world compared with 2010 because of the expected overall increase in human population by two to three billion people

and because of an increase in the wealthy fraction of this population who expect to have meat in their diet [1–5]. This will need to be achieved at the same time as meeting greenhouse gas (GHG) emission targets such as limiting global warming to 2°C or carbon neutrality as set by the Paris Agreement to the United Nations Framework Convention on Climate Change (2015) [6] as well as hitting other social and environmental targets set out in the Sustainable Development Goals (SDGs) of the United Nations [7]. However, both the future climate and the future year-to-year weather are uncertain. Thus it is difficult to design effective mitigation and adaptation strategies for policy makers who need to know what the general climate will be like as a whole and for farmers who need to manage the year-to-year risk that arises from the annual variation in the weather in the particular location that they farm.

Distinct sources of uncertainty and variability exist in environmental and agricultural systems. On the one hand, uncertainty usually reflects the amount and quality of knowledge available and follows from the assumptions and approximations and parameter inference that are necessarily involved in any modelling approach. In this paper, we consider 'climate uncertainty' in yield predictions, which results from the uncertainty associated with climate projections made with Global Climate Models (GCMs) in the Coupled Model Intercomparison Project phase 5 (CMIP5) ensemble. Besides GCM climate uncertainty, we also assess the impact of 'interannual weather variability' on yield prediction, which accounts for the interannual stochastic variability of the weather at each location. Because crop growth and development, and hence crop yields, respond nonlinearly to local weather conditions, it is important to quantify the contributions of 'climate uncertainty' and 'interannual weather variability' to uncertainty in yield prediction in the future, i.e. 2050.

The accurate assessment of crop performance under limiting factors strongly depends on the accurate representation of the relevant physiological processes governing the yield over a wide range of climatic conditions. Modelling the components of yield around flowering is key to quantifying the plant response to drought and heat stresses around flowering [8–12]. In the case of cereals such as wheat, the grain number and mass can be reduced significantly following short duration heat stress or drought events around flowering leading to substantial yields losses. This is the result of premature abortion of florets, malfunction and irreversible abortion or reduced viability of male and female reproductive organs and gametophytes, and in some cases complete male and female sterility.

With expected global warming [13], temperatures in GB will increase and summer precipitation will decrease [14,15]. Therefore, the main objective of our study is to assess how vulnerability of winter wheat production could change in future in GB climates focusing on the impact of (i) water stress over the growing season, and (ii) drought and heat stresses around flowering that affect the grain number and can substantially reduce the grain yield. Note that the effects of the expected occurrence, from natural variations in future climates [15], of some colder/drier winters and cooler/wetter summers will intrinsically be taken into account in our approach, but not the effect of waterlogging during wetter winters. Here we assess the risk of yield losses defined from stress indices (Material and methods) that compare potential yield, i.e. the yield obtained in absence of any form of stress, to the water limited yield of wheat cultivars sensitive or not to drought or heat stresses around flowering. We characterize the climate (GCM) uncertainty by using climate scenarios based on GCMs from the Coupled Model Intercomparison Project phase 5 (CMIP5) under two $CO_2$-equivalent atmospheric concentration trajectories known as Representative Concentration Pathways (RCPs) from the fifth Assessment Report (AR5) by the International Panel on Climate Change (IPCC) [16]. By contrast, we characterize the impact of the interannual weather variability from estimates of the coefficient of variation (CV) of yields that we obtained from 100 realizations of weather data per GCM scenario. Taken together, the climate uncertainty and interannual weather variability provide a snapshot of the risks to winter wheat production and the resilience of the system to climate change in GB in 2050.

Compared to previous work, this study builds upon [8] but relies on the multi-model ensemble of climate scenarios CMIP5 (from which 19 GCMs are selected) and the latest developments of the process-based wheat simulation model Sirius [17] and the LARS-WG weather generator [18]. We provide a meticulous analysis of the stress indices whose percentiles are proposed as metrics for yield vulnerability with climate change, which we believe will be useful to stakeholders from farmers to decision makers. We also concentrate on characterizing both climate uncertainty and interannual variability related to CMIP5. In this regard, our paper contrasts with previous analyses using a perturbed physics ensemble (PPE) [19,20], such as the one reported in [21], where uncertainty in climate change impacts is drawn from many model variants of a single GCM obtained by perturbing model parameters only. Our methodology is less computationally expensive and avoids the problem of GCM bias correction, whilst retaining the statistical features of climatic variability and extreme events.

# 2. Material and methods

## 2.1. Climate change scenarios and methodology outline

The wheat simulation model Sirius [17] 2018 (https://doi.org/10.5061/dryad.2v6wwpzmn) was used to model the response of winter wheat across GB to weather variability under different scenarios of climate change which were driven by greenhouse gas emissions following two Representative Concentration Pathways (RCP 4.5 and 8.5) of atmospheric $CO_2$. These pathways respectively assume an average of 487 ppm and 541 ppm of $CO_2$ in 2050, while 363.8 ppm was used for the baseline 1981–2010 climate. Based on the projections of 19 GCMs from the CMIP5 multi-model ensemble [22,23] (electronic supplementary material, table S1), local-scale climate scenarios were produced by the LARS-WG weather generator [18] (LARS-WG 6.0, https://doi.org/10.5061/dryad.3xsj3txg2). The CMIP5 experimental protocol was designed to evaluate the uncertainty on climate variability and climate change predictions resulting from the inherent modelling structural differences between GCMs. This protocol produced a multi-model dataset for both long-term (100 yr) and near-term (10–30 yr) prediction experiments from running many GCMs (59 in total) which include interactive representations of the atmosphere, ocean, land and sea ice under specified concentrations of various atmospheric constituents such as greenhouse gases. The combined use of such multi-model ensembles with a weather generator for the assessment of climate change impacts is described in [24]. In all other work, the approach is to use a single climate model in which the physics is perturbed to give an ensemble of possible outcomes (PPE) [19–21].

The 25 simulation sites were spread over GB (see map in electronic supplementary material, figure S1 and table S2 for a complete list of the sites denomination and acronyms), using for comparison purposes a single cultivar and a single soil. Our simulations were performed for a winter wheat cultivar, Mercia HD, that is sensitive to heat and drought around flowering. Cultivated since 1983, this cultivar is a British obligate (i.e. requiring vernalization) winter wheat variety with moderate to weak daylength response. Its calibration and validation in Sirius have been done previously using agronomic experiments in the UK [25–29]. Sirius demonstrated a good performance in simulating water-limited/ drought stress conditions in particular [17,26]. In the present study, we found a close agreement between Sirius (potential and water-limited) simulations for the baseline climate and experimental yield data and dates of key developmental stages for the growth of the Mercia cultivar in irrigated and unirrigated conditions during dry seasons (1993–1996) as reported in [30,31] (electronic supplementary material, figure S2). The common soil profile Hafren, which is representative of British soil types and characterized by a medium available water capacity (electronic supplementary material, table S3, Available Water Capacity, AWC = 177 mm), was used at all sites to eliminate site-specific soil effects from the analysis. Discussions on the effects of AWC variations can however be found in [28,32]. This soil profile was also assumed to be filled to the maximum available water capacity at sowing, which is consistent with real experimental conditions [30,31]. Note that the soil water deficits at flowering and maturity from our baseline Sirius simulations are comparable to the soil water deficits measured in the experimental conditions in [30,31]. Finally a single sowing date, set to 20/10 (20 October), was used in all our simulations and for all sites. For comparison and completeness, additional simulations were performed for a soil with a lower water capacity, a different cultivar and also alternative sowing dates. Results are presented and briefly discussed in the electronic supplementary material.

To give a sense of the overall trends and variability over the country, we focus our results onto countrywide spatial averaging calculated over the 25 sites. To contrast, we also analyse results in more detail from eight sub-locations split into two sets characteristic of southern and central/northern climatic conditions. Note that our results are all presented as boxplots that picture the climatic uncertainty arising from the GCMs projections. Orders of magnitude describing the direction of the crop response to climate change are discussed with respect to the medians of these boxplots, the amplitude of uncertainty being naturally represented by the extent of the boxplot's inter-quartile range and whiskers (extremes). Note that we highlight in the figures the predictions of the model HadGEM2-ES of the UK Meteorological Office using square symbols as reference points to compare with the performance of other GCMs. Depending on the variables and sites of interest, our results do not show clear and generic trends between the HadGEM2-ES predictions and the projections of other GCMs. With clear differences between GCMs apparent but no bias, simulating wheat response using all GCMs is the best way we have to express the likely variation in future climate. To estimate variation in wheat yield due to interannual variability in climate and climatic extremes, we calculated CV in wheat yields derived from simulations that used 100 years of daily weather data generated by LARS-WG at each site for each GCM, each RCP for the climate in 2050. For comparison, the baseline variability in wheat yields was also obtained from 100 sets of current weather generated by LARS-

WG (but clearly not affected by the GCMs). So, to be clear: there is no climatic uncertainty associated with the baseline, only interannual variability. The 100 realizations of current weather data for each site were based on daily observed weather data for the 1981–2010 period.

## 2.2. Impact indices

To estimate the effects of abiotic stresses on grain yield, stress-limited yields were calculated and compared with potentials. No limitation on nutrients (N, P, K) was imposed. The following three stress indices, i.e. Water Stress Index (WSI), Drought Stress Index (DSI) and Heat Stress Index (HSI), were defined as the proportions of yield lost due to water limitation during growing season, drought and heat stress around flowering, respectively:

— WSI $= (1 - Y_w/Y_p)$, where $Y_w$ is water limited yield of a cultivar tolerant to heat and drought stress around flowering, and $Y_p$ is potential yield of the same cultivar obtained in the absence of any stress;
— DSI $= (1 - Y_{wd}/Y_w)$, where $Y_{wd}$ is a water limited yield of cultivar sensitive to drought around flowering only;
— HSI $= (1 - Y_{wh}/Y_w)$, where $Y_{wh}$ is a water limited yield of cultivar sensitive to heat stress around flowering only.

Implementation details in the Sirius model are described in [11,12]. WSI represents a source capacity stress index and follows the overall water balance throughout the season. Water limitation is implemented through a moisture choking function which reduces radiation use efficiency (RUE) and soil evaporation [17]. DSI and HSI represent sink capacity indices and follow from the decrease in fertility (anthers, pollen and grain number) around flowering. Actual yield is decomposed into the product of the dry mass accumulated in ears prior to flowering, the actual number of grains per unit of ear dry mass and the actual weight of a single grain [11,12].

## 2.3. Crop simulation model

Sirius (2018) is a mechanistic wheat simulation model with a daily time step, whose detailed description and implementation are available elsewhere [11,12,17,33–36]. The model main inputs include daily weather data, a cultivar description, a soil physical description and information on crop management. Sirius combines several sub-models representing soil profile, plant phenology, water and nitrogen (N) uptake, photosynthesis and biomass production, and partitioning of photosynthases into shoot (i.e. leaf, stem, grain) and root. The effects on photosynthesis, biomass production and grain yield of abiotic stresses, such as common heat and water stresses and nutrient limitation, are taken into account, as well as drought and heat stresses during reproductive development which reduce grain number and grain size substantially affecting final grain yield. Up to a user-defined depth, soil is characterized by 5-cm layers whose physical description is expressed in terms of capacities for the soil to hold water in various states and the roots to extract water.

In summary, photosynthetic biomass is accumulated on a daily basis as the product of intercepted photosynthetically active radiation (PAR) and RUE, limited by temperature and water stress and N availability. Phyllochron, vernalization and daylength responses determine the rate of plant phenological development, from germination to maturity, in terms of primordium and leaf production, and final mainstem leaf number. Canopy development is expressed as a sequence of leaf layers whose evolution depends on a thermal time sub-model, N limitation and abiotic stresses. Throughout the crop growth period, water limitation (WSI) impedes photosynthesis, which in turn restricts biomass production and increases the rate of leaf senescence, finally reducing the grain yield.

By contrast, extreme weather events such as drought and heat stresses that occur during the reproductive phase can only perturb key stages of the inflorescence development (e.g. ovary and anthers development, premature florets abortion, irreversible abortion of male and female reproductive organs and gametophytes) and decrease significantly the primary fertile grain number. In Sirius, such effects are integrated into the drought stress (DSI) and heat stress (HSI) indices that we defined earlier. Note that heat stress and water limitation during grain filling and endosperm development can also negatively impact the potential weight of grain.

In Sirius, $CO_2$ fertilization follows from an increase in the RUE. This increase is made proportional to atmospheric $CO_2$ concentration with an increase of 30% in RUE for doubling the $CO_2$ concentration compared with the baseline of 364 ppm. This agrees with a meta-analysis of several field-scale experiments on the impacts of increased $CO_2$ concentration on crops [26,37]. Such an increase in RUE

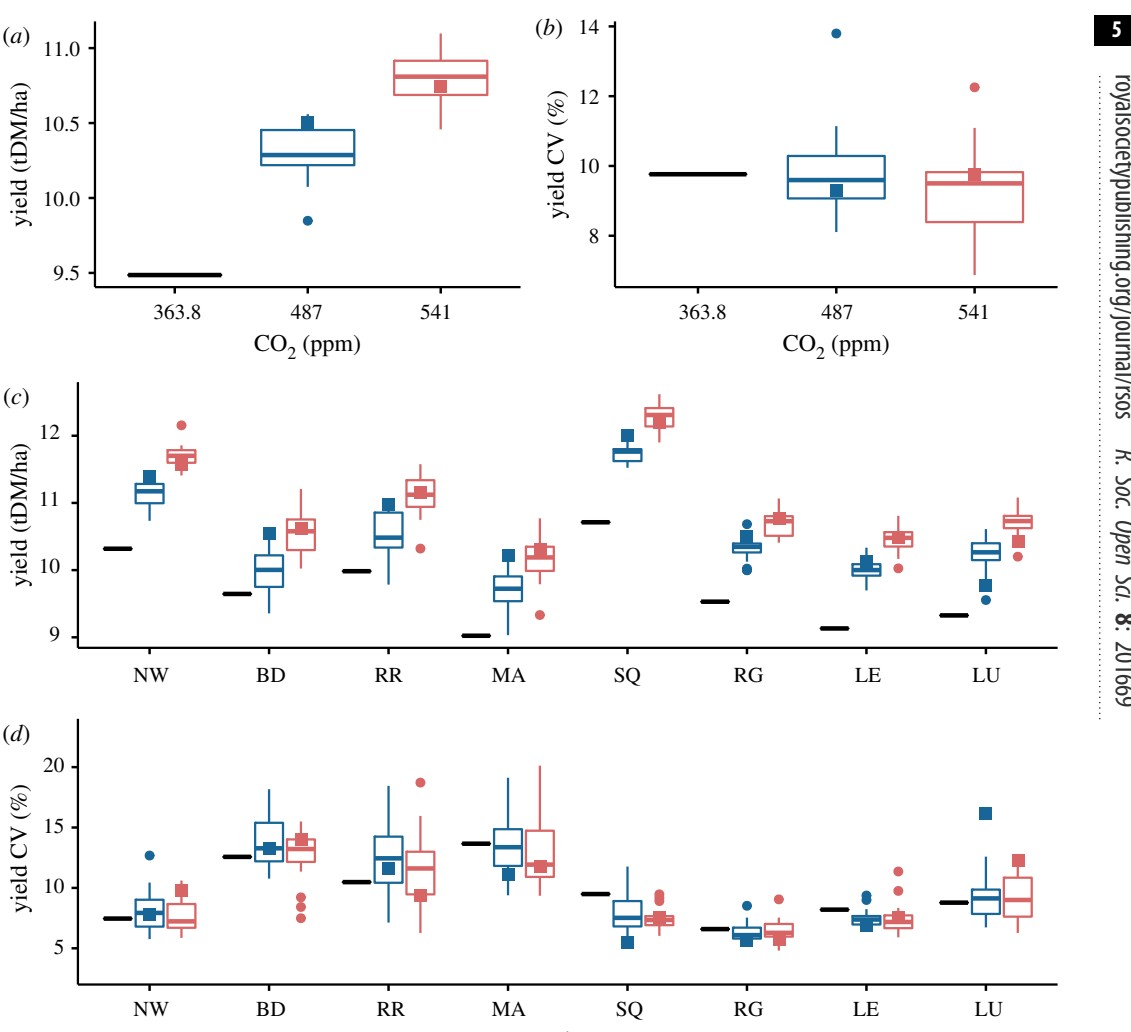

**Figure 1.** Grain yield and its coefficient of variation. (*a,b*) Spatial averages over the 25 sites (table S2) of the mean simulated grain yield and its coefficient of variation (CV) for the baseline (black line) ($[CO_2] = 363.8$ ppm) and the two 2050 scenarios RCP 4.5 ($[CO_2] = 487$ ppm) and RCP 8.5 ($[CO_2] = 541$ ppm) (respectively, blue and red box plots). (*c,d*) Site specific yield and CV at eight GB locations: NW (North Wykes), BD (Boscombe Down), RR (Rothamsted Research), MA (Marham), SQ (Saws Sennybridge), RG (Ringway), LE (Leeming), LU (Leuchars), (see map in electronic supplementary material, figure S1 and table S2 for details). Box plots are built out of the predictions for climate scenarios (each of which consists of 100 years of daily weather) derived using 19 individual GCMs from the CMIP ensemble phase 5. Box boundaries indicate the 25th and 75th percentiles, the line within the box marks the median, whiskers below and above the box indicate the 10th and 90th percentiles; dots correspond to outliers; squares specifically correspond to the predictions of the model HadGEM2-ES of the UK Meteorological Office. Simulations were performed for the wheat cultivar Mercia HD which is sensitive to heat and drought stress around flowering, and a single soil with medium available water content (AWC = 177), the sowing date being 20/10.

is unlikely to continue to increase linearly with $CO_2$ but our RCP concentrations (487, 541 ppm) fall within the range of linear increase found in the meta-analysis. The Sirius performance with respect to temperature and $CO_2$ increases under diverse climates can be found in several crop modelling validation studies of the Agricultural Model Intercomparison and Improvement Project (AgMIP) [38–40].

# 3. Results and discussion

## 3.1. Assessment of future yield

We focus on the simulations performed for the winter wheat cultivar, Mercia HD, that is sensitive to heat and drought around flowering. Because of $CO_2$ fertilization, yields increased on average over the whole

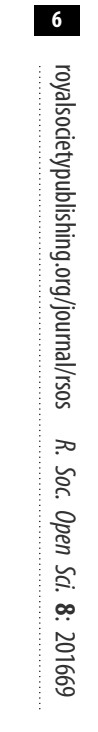

**Figure 2.** Key weather variables. Cumulative values integrated over the season period. (*a–c*) Spatial averages over the 25 sites and (*d–f*) site specific local prediction for the temperature sum (°C d), total rainfall (mm) and total intercepted solar radiation (MJ m$^{-2}$). The nomenclature of the boxplots is explained in figure 1. Cultivar: Mercia HD; Soil: Hafren (AWC = 177); Sowing date: 20/10.

country by 8.9% and 14.3% compared to a baseline for the RCPs 4.5 and 8.5, respectively (figure 1*a*). This rise in productivity is modest, accounting for no more than 1.5 t/ha but is comparable to predictions for different, less sensitive wheat cultivars across the rest of Europe [8] and in GB (electronic supplementary material, figure S4 and table S4)). Note that, in the absence of $CO_2$ fertilization, yields of the cultivar Mercia would drop by about 0.25 t/ha on average (electronic supplementary material, figure S3).

Locally, mean yields vary from 8 to 12 t/ha but are noticeably homogeneous across sites (figure 1*c*). The differences, however, can be explained in terms of well understood trends in climatic variation across GB such as the decrease in mean temperature towards the north of the country (figure 2*d*), or the decrease in mean precipitation towards the east of the country (figure 2*e*), while radiation varies slightly with a west to east gradient between the sites (figure 2*f*). More precisely, comparing figure 1*c* and figure 3*c*, we see that yield variation between sites follows that of the seasonal evapotranspiration (ET). In our case, the differences in ET directly reflect the level of water deficit between locations, which in this study, are the result of the weather variability only, the soil being assumed the same at each site (figure 3*d*). The higher the water deficit the lower the ET, and correspondingly, the yield. Note however the exception of the Scottish site LU, which has both higher ET and yield than the sites nearby RG and LE. This is explained by the relative higher radiation at site LU (figure 2*f*). Following the British weather pattern described above, the soil water deficit exhibits a clear division between the southern sites (NW, BD, RR, MA) more prone to water stress, and the northern sites (SQ, RG, LE, LU) less sensitive to water stress. We note that the soil water deficit increases modestly under the RCP 4.5 scenario and varies non-monotonically with the atmospheric $CO_2$ concentration (figure 3*b,d*). This can

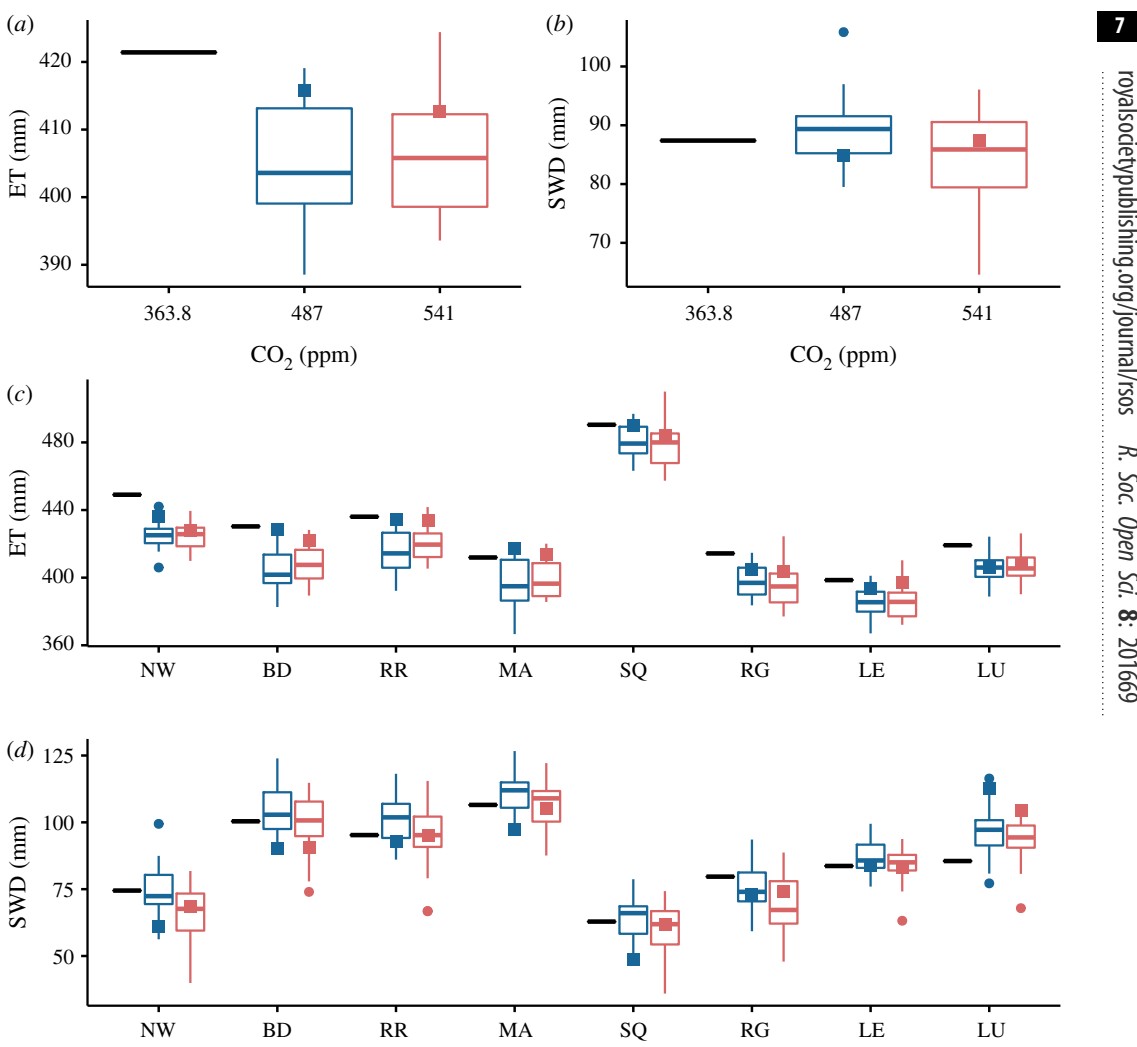

**Figure 3.** Key water balance variables at anthesis. (*a,b*) Spatial averages over the 25 sites and (*c,d*) site specific local prediction for the seasonal evapotranspiration (ET), and soil water deficit (SWD) at flowering. Nomenclature explained in figure 1. Cultivar: Mercia HD; Soil: Hafren (AWC = 177); Sowing date: 20/10.

be explained by an increase in precipitation and a reduction in radiation in the RCP 8.5 scenario compared to the RCP 4.5 (figure 2*b,c*).

The coefficient of variation of wheat yield in figure 1*b,d* provides a metric for comparing the baseline interannual weather variability throughout 1980–2010 with that found in 2050. Averaged over the country, the summary statistics of 2050 simulations show that the medians of CV drop by about one percent compared to the baseline (from 10.1% for the baseline to 9.3% for the two RCPs). Hence climate change modestly reduces the yield interannual variability because of the reduction in the ET. This follows from a shorter growing season and the associated reduction in water stress. This trend is also confirmed locally at each site. Looking more closely however, the yield coefficient of variation in figure 1*d* parallels the water deficit in figure 3*d*. This suggests that water deficit locally controls the level of interannual yield variability as higher level of water deficit increases yield variability. Note however that exceptions to this interannual variability reduction exist. The site RR has a substantial increase in yield CV in 2050 compared to baseline. Our study in effect shows that the climate uncertainty is significant with respect to the value of yield CV, which could reach a 50% increase compared to baseline and even exceeds 15% for the projection of the most extreme GCM in 2050. This is important. Indeed, following the common interpretation of CV as a yield stability indicator (e.g. [41]) our predictions demonstrate that the yield of winter wheat could then become substantially unstable by 2050, in the southeast of GB, even from a relatively modest increase in atmospheric $CO_2$.

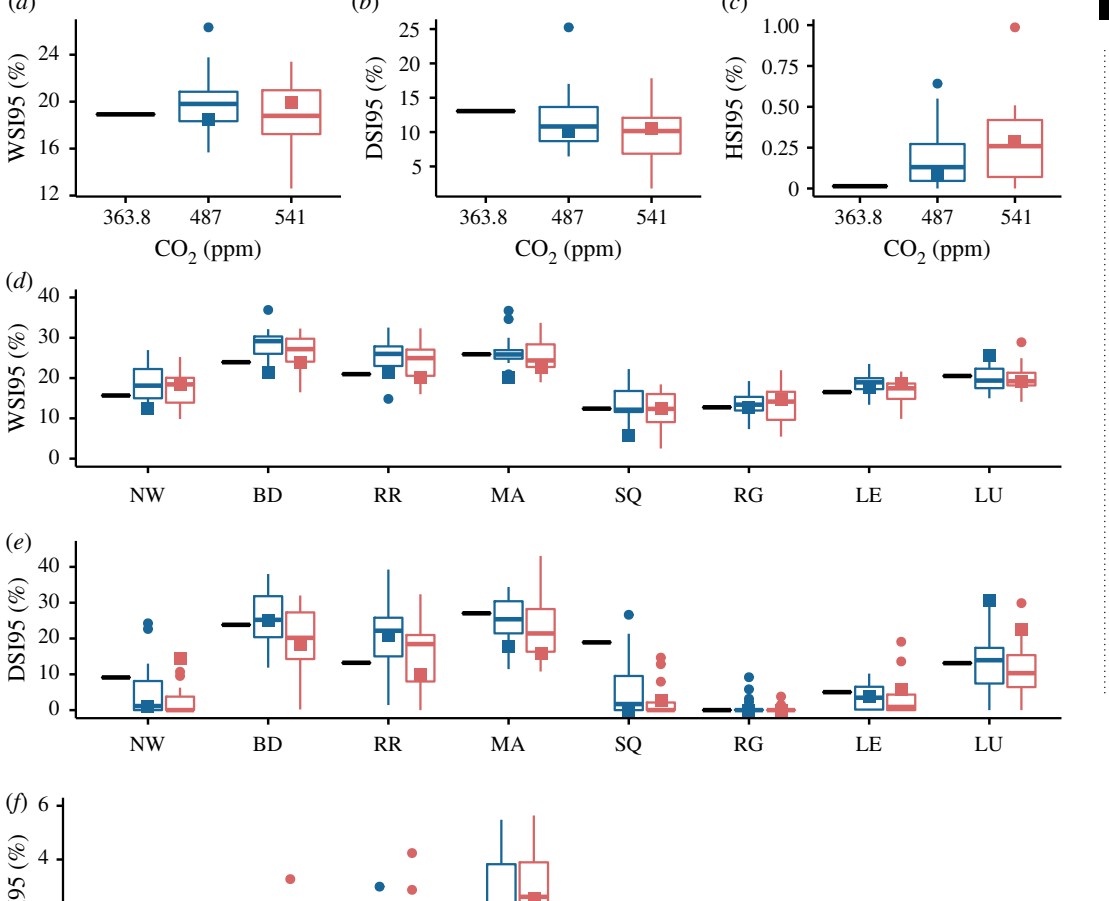

**Figure 4.** Environmental stresses. (*a–c*) Spatial averages over the 25 sites and (*d–f*) site specific local prediction for the 95-percentiles of the water, drought and heat stress indices WSI, DSI, HSI. The stress indices are defined as $\text{WSI} = (1 - Y_w/Y_p)$, $\text{DSI} = (1 - Y_{wd}/Y_w)$, $\text{HSI} = (1 - Y_{wh}/Y_w)$, where $Y_w$ is water limited yield while $Y_p$ is potential yield of a cultivar tolerant to heat and drought stress around flowering, $Y_{wd}$ is a yield of cultivar sensitive to drought around flowering, $Y_{wh}$ is a yield of cultivar sensitive to heat stress around flowering (see Methods for more details). Nomenclature explained in figure 1. Cultivar: Mercia HD; Soil: Hafren (AWC = 177); Sowing date: 20/10.

## 3.2. Environmental stresses and yield losses

The geographical variation in water deficit can be expressed succinctly by the WSI, DSI and HSI indices (figure 4). We concentrate on the 95-percentile of these indices, which gives estimates of the yield losses due to extreme weather events. For instance, the index WSI95 represents the minimum level of yield losses that has a 5% chance of occurrence (i.e. Prob(WSI > WSI95) = 0.05), that is the yield loss to occur once in every 20 years. The yield losses from seasonal water stress (WSI95) is much greater than heat stress around flowering (HSI95) over the whole country, while yield losses brought out by drought stress (DSI95) around flowering will account for about 10% of the total yield loss resulting from seasonal water stress (figure 4*a–c*). Numerically, the medians of WSI95 and DSI95 for the two RCPs differ very little from the baseline and remain about 20% and 10% on average. The general trend then is on of a levelling off and a modest reduction in the seasonal water stress and the drought stress around flowering compared to baseline, respectively. However, we also see clearly (figure 4*c*) that heat stress around flowering is an emerging threat. Such a heat stress represents a new potential threat for winter wheat production in GB thus increasing its vulnerability to global warming, as has been suggested for the rest of Europe [8].

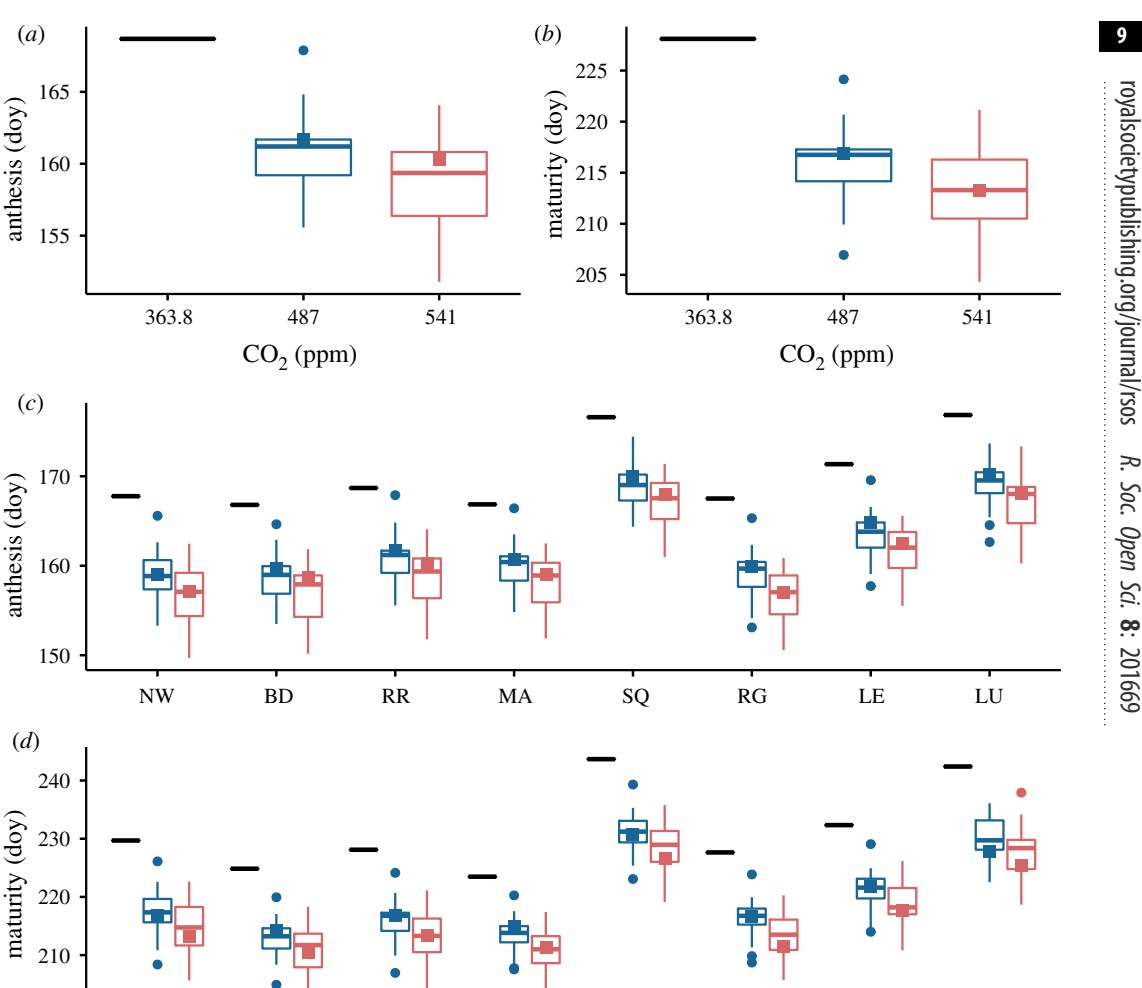

**Figure 5.** Anthesis and maturity dates. For all the scenarios considered in our study, our simulations predict a shift in phenology due to climate change. Dates are expressed in day of year (doy). Nomenclature explained in figure 1. Cultivar: Mercia HD; Soil: Hafren (AWC = 177); Sowing date: 20/10.

Locally, however, the levelling off and modest reduction in water and drought stresses observed at the scale of the country is misleading because most of the wheat production is concentrated in southern Britain. Geographically, with a clear eastwards gradient towards drier climate, figure 4*d,e* shows that the local trends of the seasonal water stress and drought stress around flowering follow those of the soil water deficit, and in turn, the north/south dichotomy discussed above. At some sites in the south of the country (e.g. NW, BD, RR), yield losses from seasonal water stress could rise modestly by a few percents due to climate change. Note finally that losses due to heat stress are negligible for most sites (figure 4*f*).

Interestingly, the climatic uncertainty caused by the variations in the GCM predictions for drought and heat stresses around flowering is much larger for sites located in the southeast like RR compared to the rest of the country. Furthermore, at specific sites such as RR, our simulations predict a significant increase in the drought stress contribution to yield losses, which is reflected in the yield CV (figure 1). Nevertheless, in contrast to the south of Europe for instance [8], the drought and heat stresses around flowering will remain relatively small compared to overall water stress. Compared to baseline, this comes about because anthesis and maturity come forward by between 8 and 12 days on average due to accelerated development resulting from the increase in mean temperature (figures 2 and 5), respectively shifting flowering and harvest periods to early June and August. Heat stress around flowering is almost always avoided therefore. For the RCP 8.5 pathway, the anthesis and maturity dates occur still earlier (2–3 days) than for the RCP 4.5 pathway. We note that such a phenology shift is a remarkable natural adaptation of wheat for avoiding drought and heat stresses around flowering.

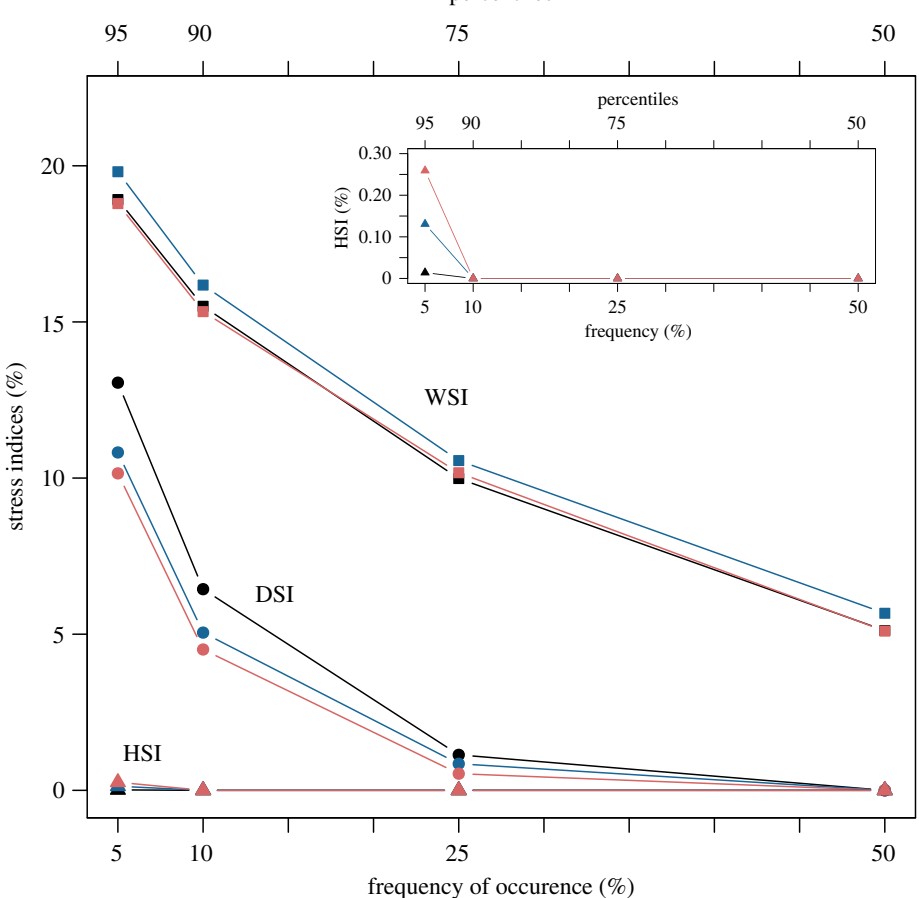

**Figure 6.** Yield loss levels due to water stress versus frequency of occurrence. The frequencies correspond to the probabilities Prob(WSI > WSI$_{percentiles}$) associated with the 50, 75, 90, 95–percentiles of WSI. The data points represent the spatially averaged value of the medians of these four percentiles of WSI, as seen in the boxplots in figure 4. Projections for 2050 show that yield losses remain roughly the same compared to baseline. Nomenclature explained in figure 1. Cultivar: Mercia HD; Soil: Hafren (AWC = 177); Sowing date: 20/10.

## 4. Concluding remarks

Wheat is one of the most important crops cultivated in both GB, and worldwide. At the scale of the country, our results demonstrate that the seasonal water stress and the drought stress around flowering will remain about the same or reduce slightly compared to a 1980–2010 baseline. Heat stress around flowering, however, will begin to become a problem, but remain substantially lower than water stress. Overall, despite climate change, winter wheat production will continue to be favourable in GB, because of the benefits of $CO_2$ fertilization and the shift in flowering phenology to earlier in the year.

The overall seasonal water stress level (WSI) is an estimate of the relative magnitude of the yield gap. We can summarize our results , by plotting this stress level as a function of frequency of occurrence driven by the weather variability defined by 50, 75, 90 and 95-percentiles of WSI (figure 6). We recall that these percentiles represent yield losses which are expected to occur every 2, 4, 10 and 20 years on average, respectively. Figure 6 shows that yield losses remain bounded between 5% and 20%, future losses under the two RCP scenarios staying close to baseline. In other words, the impact of extreme but rare events associated with the weather variability on seasonal water stress will stay limited and of the same order of magnitude as the last three decades on average.

These percentiles of WSI can be interpreted as yield resilience/risk indicators across spatial scales. They apply from the country scale as pictured in figure 6 to the site-specific scale as shown in figure 4. As such, they can be valuable to both the decision maker and the farmer because they provide estimates of the level of yield losses and the frequency of a level of loss due to weather variability. We

suggest that the 95-percentile estimate might be relevant to the decision maker interested in long-term planning, while the 50,75-percentile estimates might be more informative to the farmer interested in shorter term risk assessment. Taking into account climate uncertainty embedded in the GCMs projections, we point out that our numerical results give a lower bound of the future amplitude of yield losses due to water stress. For instance, key factors such as saturation of the $CO_2$ fertilization effect, elevated night temperatures or an indirect reduction in the available water in soil (caused by compaction or loss in organic matter due to warming) will eventually impact negatively on yield and increase the risk and magnitude of yield losses.

Finally, we emphasize that our results also indicate that the crop response to climate change can be subtle. Regionally, the most striking feature that we observe is the clear divide between the southeastern of England (i.e. the southeast and east of England regions), where most of cereal crops are grown and whose CV exceed 10%, and the rest of the country, where precipitation is larger and where CV barely reach 10%. The southeast is also characterized by greater GCM uncertainty (figure 1). In particular, we notice a significant increase in the median of the coefficient of variation in the CGM variability for RR (figure 1*d*). This strongly suggests that the east/southeast of England is most likely to suffer from greater uncertainty in terms of crop productivity and sensitivity to extreme events in the future. Consequently yields could become unstable and production more vulnerable to climate change, especially in places where the soil available water content is limited. Our numerical results show that yield stability, without irrigation management, could be achieved primarily with the right choice of cultivar and sowing dates (electronic supplementary material, figures S4, S6 and table S4).

Our simulations demonstrate that the local level of ET is a key variable determining both the level of yield and the water deficit. The fact that climate change negatively impacts the level of ET suggests that the moderate yield gain caused by $CO_2$ fertilization may be lost beyond 2050 in the east and southeast. These words of caution are consistent with previous work [21] and recent large scale simulations for unmitigated climate change [42]. Because $CO_2$ fertilization appears to be the chief mitigating factor, a better understanding and better modelling [43] of this process will be key in increasing the reliability and reducing the uncertainty of any future projections of yield of winter wheat, and its resilience and stability in GB with climate change.

Data accessibility. This article has no additional data.

Authors' contributions. T.P. was responsible for analysing the results and drafting the manuscript. A.P.W. brought perspective on the analysis of the results, and together with M.A.S. and others wrote the CROP-NET project grant. M.A.S. and N.S. were responsible for model development and simulations. M.A.S. conceived and led the modelling approach. All authors contributed to writing and reviewed the manuscript and gave final approval for publication.

Competing interests. We declare we have no competing interests.

Funding. Rothamsted Research receives grant-aided support from the Biotechnology and Biological Sciences Research Council (BBSRC) through Designing Future Wheat [BB/P016855/1] and for the Soils to Nutrition programme [BBS/E/C/000I0330] and Achieving Sustainable Agricultural Systems [NE/N018125/1]. The authors acknowledge support from NERC for the CROP-NET project [NE/S016821/1].

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
