## [Peer Review File · Royal Society Open Science]

Review History

RSOS-201669.R0 (Original submission)

Review form: Reviewer 1

Is the manuscript scientifically sound in its present form?

Yes

Are the interpretations and conclusions justified by the results?

Yes

Is the language acceptable?

Yes

Do you have any ethical concerns with this paper?

No

Have you any concerns about statistical analyses in this paper?

No

Recommendation?

Accept as is

Comments to the Author(s)

I have reviewed a R1 manuscript. The authors replied to the feedback of the reviewers and I have no additional questions.

Review form: Reviewer 2**Is the manuscript scientifically sound in its present form?**

No

Are the interpretations and conclusions justified by the results?

No

Is the language acceptable?

Yes

Do you have any ethical concerns with this paper?

No

Have you any concerns about statistical analyses in this paper?

No

Recommendation?

Major revision is needed (please make suggestions in comments)

Comments to the Author(s)

The responses of the authors to major critics of the previous three referees are not sufficient. I think an appropriate response to the reviewers requests needs another major revision. Title, abstract, research question raised and methods chosen should follow a recognizable logic. This is currently not the case and requests a major overhaul of the study and the manuscript.

Response to Ref #1

The handling of drought stress in the manuscript is still confusing. The authors should simply adjust terms. They consider a mean season drought stress and a drought stress around anthesis. Just use one abbreviation with two indices and don't confuse the reader by the hopeless attempt to introduce a new terminology.

Response to Ref #2

The authors still suggest in their modified title they would address the future winter wheat cropping in Great Britain. The authors downplay the meaning of the title for the reviewer but not for the reader. The reader still expects a comprehensive regional study.

The focus of the paper is unclear: 'This paper will primarily focus on estimating the level of this climate uncertainty and weather variability on crop yield in the future'.

What is meant with, "estimating the level of this climate uncertainty"? I cannot find the answer in the abstract of the paper or elsewhere.

The study has value only for the test sites where real soil conditions were used as basis. At all other location the reader is still left with the question, how will the outcome be sustained when the actual soil instead of the generalised one is taken into account?

For a source of variability study with a representative simulation of the soil x climate interactions for Great Britain the authors might use the following analysis as an orientation (reviewer not among the authors):

Hattermann, F. F., Vetter, T., Breuer, L., Su, B., Daggupati, P., Donnelly, C., Fekete, B., Flörke, F., Gosling, S. N., Hoffmann, P., Liersch, S., Masaki, Y., Motovilov, Y., Müller, C., Samaniego, L., Stacke, T., Wada, Y., Yang, T., Krysanova, V. (2018): Sources of uncertainty in hydrological climate impact assessment: a cross-scale study. - *Environmental Research Letters*, 13, 1, 015006. <https://doi.org/10.1088/1748-9326/aa9938>

Response to Ref 3

Ref 3 has the same concern as Ref 2 in respect to the representative value of the study. I share these concerns and think the authors current reply is not sufficient in this respect.

Instead of keeping the soil profiles constant I would recommend to vary them as suggested by Ref 3 and apply an analysis of variance similar to the one above.

Reply to the Ref 3 Page 3 line 79 comment. The authors need to explain better their motivation for combining the GCM ensemble outcome with one weather generator. It would be interesting to see the different effects of an impact study that a) leaves the GCM stochastic more or less untreated and corrects for the mean bias only and b) that combines the GCM outcome with a weather generator.

Decision letter (RSOS-201669.R0)

Dear Dr Putelat

On behalf of the Editors, we are pleased to inform you that your Manuscript RSOS-201669 "Impacts of climate change on winter wheat in Great Britain" has been accepted for publication in Royal Society Open Science subject to minor revision in accordance with the referees' reports. Please find the referees' comments along with any feedback from the Editors below my signature.

Please submit your revised manuscript and required files (see below) no later than 7 days from today's (ie 17-Feb-2021) date. Note: the ScholarOne system will 'lock' if submission of the revision is attempted 7 or more days after the deadline. If you do not think you will be able to meet this deadline please contact the editorial office immediately.

Best regards,
Lianne Parkhouse

Editorial Coordinator
Royal Society Open Science
openscience@royalsociety.org

on behalf of Dr Yhasmin Mendes de Moura (Associate Editor) and Pete Smith (Subject Editor)
openscience@royalsociety.org

Subject Editor comments to the authors:

The Associate Editor recommends acceptance and I am happy to follow this recommendation, but I note a number of critical comments from Reviewer 2, which I think it would be useful to address when submitting your final version. Most comments can be addressed through rewording and change of emphasis, which will preempt similar criticisms from the readers, so I urge you to consider them carefully.

Best wishes,
Pete Smith

Associate Editor Comments to Author (Dr Yhasmin Mendes de Moura):

Dear author's,

I am delighted to recommend this manuscript for publication at Open Science Royal Society.

Congratulations!

Best regards,
Reviewer comments to Author:

Reviewer: 1
Comments to the Author(s)

I have reviewed a R1 manuscript. The authors replied to the feedback of the reviewers and I have no additional questions.

Reviewer: 2
Comments to the Author(s)

The responses of the authors to major critics of the previous three referees are not sufficient. I think an appropriate response to the reviewers requests needs another major revision. Title, abstract, research question raised and methods chosen should follow a recognizable logic. This is currently not the case and requests a major overhaul of the study and the manuscript.

Response to Ref #1

The handling of drought stress in the manuscript is still confusing. The authors should simply adjust terms. They consider a mean season drought stress and a drought stress around anthesis. Just use one abbreviation with two indices and don't confuse the reader by the hopeless attempt to introduce a new terminology.

Response to Ref #2

The authors still suggest in their modified title they would address the future winter wheat cropping in Great Britain. The authors downplay the meaning of the title for the reviewer but not for the reader. The reader still expects a comprehensive regional study.

The focus of the paper is unclear: 'This paper will primarily focus on estimating the level of this climate uncertainty and weather variability on crop yield in the future'.

What is meant with, "estimating the level of this climate uncertainty"? I cannot find the answer in the abstract of the paper or elsewhere.

The study has value only for the test sites where real soil conditions were used as basis. At all other location the reader is still left with the question, how will the outcome be sustained when the actual soil instead of the generalised one is taken into account?

For a source of variability study with a representative simulation of the soil x climate interactions for Great Britain the authors might use the following analysis as an orientation (reviewer not among the authors):

Hattermann, F. F., Vetter, T., Breuer, L., Su, B., Daggupati, P., Donnelly, C., Fekete, B., Flörke, F., Gosling, S. N., Hoffmann, P., Liersch, S., Masaki, Y., Motovilov, Y., Müller, C., Samaniego, L., Stacke, T., Wada, Y., Yang, T., Krysanova, V. (2018): Sources of uncertainty in hydrological climate impact assessment: a cross-scale study. - *Environmental Research Letters*, 13, 1, 015006. <https://doi.org/10.1088/1748-9326/aa9938>

Response to Ref 3

Ref 3 has the same concern as Ref 2 in respect to the representative value of the study. I share these concerns and think the authors current reply is not sufficient in this respect.

Instead of keeping the soil profiles constant I would recommend to vary them as suggested by Ref 3 and apply an analysis of variance similar to the one above.

Reply to the Ref 3 Page 3 line 79 comment. The authors need to explain better their motivation for combining the GCM ensemble outcome with one weather generator. It would be interesting to see the different effects of an impact study that a) leaves the GCM stochastic more or less untreated and corrects for the mean bias only and b) that combines the GCM outcome with a weather generator.

===PREPARING YOUR MANUSCRIPT===

If you have been asked to revise the written English in your submission as a condition of publication, you must do so, and you are expected to provide evidence that you have received language editing support. The journal would prefer that you use a professional language editing service and provide a certificate of editing, but a signed letter from a colleague who is a native

speaker of English is acceptable. Note the journal has arranged a number of discounts for authors using professional language editing services (<https://royalsociety.org/journals/authors/benefits/language-editing/>).

===PREPARING YOUR REVISION IN SCHOLARONE===

-- If you have uploaded ESM files, please ensure you follow the guidance at <https://royalsociety.org/journals/authors/author-guidelines/#supplementary-material> to include a suitable title and informative caption. An example of appropriate titling and captioning

may be found at https://figshare.com/articles/Table_S2_from_Is_there_a_trade-off_between_peak_performance_and_performance_breadth_across_temperatures_for_aerobic_sc_ope_in_teleost_fishes_/3843624.

Author's Response to Decision Letter for (RSOS-201669.R0)

See Appendix A.

Decision letter (RSOS-201669.R1)

Dear Dr Putelat,

I am pleased to inform you that your manuscript entitled "Local impacts of climate change on winter wheat in Great Britain" is now accepted for publication in Royal Society Open Science.

Please accept our apologies for the delay here: a miscommunication between editorial office and the editors regrettably caused some confusion.

If you have not already done so, please ensure that you send to the editorial office an editable version of your accepted manuscript, and individual files for each figure and table included in your manuscript. You can send these in a zip folder if more convenient. Failure to provide these files may delay the processing of your proof. Y

Please see the Royal Society Publishing guidance on how you may share your accepted author manuscript at <https://royalsociety.org/journals/ethics-policies/media-embargo/>. After publication, some additional ways to effectively promote your article can also be found here

<https://royalsociety.org/blog/2020/07/promoting-your-latest-paper-and-tracking-your-results/>.

on behalf of Dr Yhasmin Mendes de Moura (Associate Editor) and Pete Smith (Subject Editor)
openscience@royalsociety.org

Appendix A

Response to Reviewer 2.

Title: Local impacts of climate change on winter wheat in Great Britain

Authors: T. Putelat, A.P. Whitmore, N. Senapati, M.A. Semenov

Journal: RSOS

We thank the reviewer for his or her review, comments and observations. We have revised accordingly.

Response to Ref #1.

The handling of drought stress in the manuscript is still confusing. The authors should simply adjust terms. They consider a mean season drought stress and a drought stress around anthesis. Just use one abbreviation with two indices and don't confuse the reader by the hopeless attempt to introduce a new terminology.

WSI is a seasonal stress from water limitation for a cultivar tolerant to heat and drought stress around anthesis. This type of stress can be simulated by many crop models. However, not all crop models incorporate responses to heat and drought stress around anthesis, but Sirius does. Therefore, we decided to analyse two additional stress indices, DSI and HSI, which simulate the specific reduction in yield due to decreases in grain number and grain size as a result of drought or heat stress around anthesis. By doing so, we analyse how these stress indices will change under climate change. This type of analysis is critical for understanding severity of impacts of climate change on wheat. We have carefully revised the abstract and the main text to avoid any ambiguity between definition and usage of terms for the various stresses. Please, note, that the original R1 was satisfied with the revised manuscript.

Response to Ref #2.

The authors still suggest in their modified title they would address the future winter wheat cropping in Great Britain. The authors downplay the meaning of the title for the reviewer but not for the reader. The reader still expects a comprehensive regional study.

The title has been changed to "Local impacts of climate change on winter wheat in Great Britain".

The focus of the paper is unclear: 'This paper will primarily focus on estimating the level of this climate uncertainty and weather variability on crop yield in the future'. What is meant with, "estimating the level of this climate uncertainty"? I cannot find the answer in the abstract of the paper or elsewhere.

The focus of the paper is an investigation of the impacts of climate change on winter wheat in Great Britain (as in the title). To clarify the treatment of uncertainty, we revised the paragraph (page 2, lines 25-31) as:

“In this paper, we consider “climate uncertainty” in yield predictions, which results from the uncertainty associated with climate projections from Global Climate Models (GCMs) in the CMIP5 ensemble. Besides GCM climate uncertainty, we also assess the impact of “interannual weather variability” on yield prediction, which accounts for the interannual stochastic variability of the weather at each location. Because crop growth and development, and hence crop yields, respond nonlinearly to local weather conditions, it is important to quantify the contributions of “climate uncertainty” and “interannual weather variability” to uncertainty in yield prediction in the future, i.e. 2050.”

We have also carefully revised the abstract.

The study has value only for the test sites where real soil conditions were used as basis. At all other location the reader is still left with the question, how will the outcome be sustained when the actual soil instead of the generalised one is taken into account? For a source of variability study with a representative simulation of the soil x climate interactions for Great Britain the authors might use the following analysis as an orientation (reviewer not among the authors): Hattermann, F. F., Vetter, T., Breuer, L., Su, B., Daggupati, P., Donnelly, C., Fekete, B., Flörke, F., Gosling, S. N., Hoffmann, P., Liersch, S., Masaki, Y., Motovilov, Y., Müller, C., Samaniego, L., Stacke, T., Wada, Y., Yang, T., Krysanova, V. (2018): Sources of uncertainty in hydrological climate impact assessment: a cross-scale study. - Environmental Research Letters, 13, 1, 015006. <https://doi.org/10.1088/1748-9326/aa9938>

We have already addressed the question of using variable soils by adding simulations for a “poor” soil with AWC =127 mm, which we reported in SI. This was our response to the original R2:

“However, to tackle this issue of soil variability, we have performed further simulation with a different soil with a low available water content (AWC=127 mm). The results are presented in the S.I. Fig. 5 and briefly analysed in the SI document. We find that the reduction of the soil water capacity does not change the trends that we have described in the original text of our paper. However, as one can expect, the reduction of AWC is accompanied by a reduction of yield (that can reach 1 t/ha) and an increase in interannual variability (that can reach 15% instead of the 10% we find for the original soil which is less sensitive to water limitation). We also refer to a companion paper (Harkness et al., 2020) where soil variability is studied in more detail.”

We believe that investigation of impacts of climate change on wheat in Great Britain is clearly presented in the revised manuscript. In our study, we focused on analysis of “climate uncertainty” and “interannual weather variability” using a single crop model with 19 CMs and 2 RCPs. We believe that the manuscript will have little benefit from the additional analysis of soil x climate interactions (similar to Hattermann et al 2018 as suggested by R2). Moreover, the key element of the Hattermann et al 2018 paper was the comparison/uncertainty between 13 hydrological models with 5 GCMs and 3 RCPs, which is not the case in our study.

Response to Ref #3.

Ref 3 has the same concern as Ref 2 in respect to the representative value of the study. I share these concerns and think the authors current reply is not sufficient in this respect. Instead of keeping the soil profiles constant I would recommend to vary them as suggested by Ref 3 and apply an analysis of variance similar to the one above.

By selecting a single soil profile for all locations (AWC=177 mm or AWC=127 mm) we focused on investigation of the impact of climate change on wheat. This is a standard approach which allows to exclude the effect of soil variation, and to concentrate on studying the effects of changed climate. This was our response to the original R3:

“The revised version of our paper is now more comprehensive and includes in the SI document new simulations results for 1) an additional soil type with a low AWC, 2) an additional cultivar, 3) additional sowing dates and 4) simulations without the CO₂ fertilization effect. The new results confirm our initial conclusions”.

Reply to the Ref 3 Page 3 line 79 comment. The authors need to explain better their motivation for combining the GCM ensemble outcome with one weather generator. It would be interesting to see the different effects of an impact study that a) leaves the GCM stochastic more or less untreated and corrects for the mean bias only and b) that combines the GCM outcome with a weather generator.

GCMs from the CMIP5 ensemble do not reproduce well interannual weather variability due to their coarse spatial and temporal resolution. Therefore, the use of the raw (bias-corrected) outputs from GCMs with process-based crop models, which respond to weather variability in non-linear way, is not recommended. We use a single WG as a mean of downscaling climate projections from GCMs to a local scale. As we explained in our response to the original R3, the use of many WGs should make little difference to conclusions of our study. This was our original response to the original R3:

“WG is not a bottleneck, because WG does not predict future climates. All WGs are designed to reproduce baseline climate accurately. Therefore, the use of two weather generators WG1 and WG2 with a single GCM should produce very similar climate scenarios for the future. But two GCMs, GCM1 and GCM2, will produce very different climate projections and after downscaling very different climate scenarios. That is why in our study we estimated uncertainty due to different GCMs from CMIP5.”

REVISED ABSTRACT.

Under future CMIP5 climate change scenarios for 2050 an increase in wheat yield of about 10% is predicted in Great Britain as a result of the combined effect of CO₂ fertilization and a shift in phenology. Compared to the present day, crops escape increases in the climate impacts of drought and heat stresses on grain yield by developing before these stresses can occur. In the future, yield losses from water stress over growing season will remain about the same across Great Britain with losses reaching around 20% of potential yield, whilst losses from drought around flowering will decrease and account for about 9% of water limited yield. Yield losses from heat stress around flowering will remain negligible in the future. These conclusions are drawn from a modelling study based on the response of the Sirius wheat simulation model to local-scale 2050-climate scenarios derived from 19 Global Climate Models from the CMIP5 ensemble at 25 locations representing wheat growing areas in Great Britain. However, depending on susceptibility to water stress, substantial interannual yield variation between locations is predicted, in some cases suggesting low wheat yield stability. For this reason, local-scale studies should be performed to evaluate uncertainties in yield prediction related to future weather patterns.